# Near-Surface Soil Moisture Characterization in Mississippi's Highway Slopes Using Machine Learning Methods and UAV-Captured Infrared and Optical Images

Rakesh Salunke [1,*], Masoud Nobahar [1], Omer Emad Alzeghoul [1], Sadik Khan [1], Ian La Cour [2] and Farshad Amini [1]

1    Department of Civil and Environmental Engineering, Jackson State University, 1400 J.R. Lynch Street, JSU Box 17068, Jackson, MS 39217, USA
2    Mississippi Department of Transportation (MDOT), Materials Division, Geotechnical Branch, 412 E. Woodrow Wilson Ave., Jackson, MS 39216, USA
*    Correspondence: rakesh.salunke@students.jsums.edu; Tel.: +1-205-292-4667

**Abstract:** Near-surface soil moisture content variation is a major factor in the frequent shallow slope failures observed on Mississippi's highway slopes built on expansive clay. Soil moisture content variation is monitored generally through borehole sensors in highway embankments and slopes. This point monitoring method lacks spatial resolution, and the sensors are susceptible to premature failure due to wear and tear. In contrast, Unmanned/Uncrewed Aerial Vehicles (UAVs) have higher spatial and temporal resolutions that enable more efficient monitoring of site conditions, including soil moisture variation. The current study focused on developing two methods to predict soil moisture content ($\theta$) using UAV-captured optical and thermal combined with machine learning and statistical modeling. The first method used Red, Green, and Blue (RGB) color values from UAV-captured optical images to predict $\theta$. Support Vector Machine for Regression (SVR), Extreme Gradient Boosting (XGB), and Multiple Linear Regression (MLR) models were trained and evaluated for predicting $\theta$ from RGB values. The XGB model and MLR model outperformed the SVR model in predicting soil moisture content from RGB values. The $R^2$ values for the XGB and MLR models were >0.9 for predicting soil moisture when compared to SVR ($R^2 = 0.25$). The Root Mean Square Error (RMSE) for XGB, SVR, and MLR were 0.009, 0.025, and 0.01, respectively, for the test dataset, affirming that XGB was the best-performing model among the three models evaluated, followed by MLR and SVR. The better-performing XGB and MLR models were further validated by predicting soil moisture using unseen input data, and they provided good prediction results. The second method used Diurnal Land Surface Temperature variation ($\Delta$LST) from UAV-captured Thermal Infrared (TIR) images to predict $\theta$. TIR images of vegetation-covered areas and bare ground areas of the highway embankment side slopes were processed to extract $\Delta$LST amplitudes. The underlying relationship between soil surface thermal inertia and moisture content variation was utilized to develop a predictive model. The resulting single-parameter power curve fit model accurately predicted soil moisture from $\Delta$LST, especially in vegetation-covered areas. The power curve fit model was further validated on previously unseen TIR, and it predicted $\theta$ with an accuracy of RMSE = 0.0273, indicating good prediction performance. The study was conducted on a field scale and not in a controlled environment, which aids in the generalizability of the developed predictive models.

**Keywords:** soil moisture content; highway embankment; UAV; Thermal Infrared (TIR); machine learning; XGBoost; support vector regression; multiple linear regression

## 1. Introduction

Soil moisture content variation directly impacts shear strength, volume change, and crack formation within geo-structures such as highway slopes and embankments. Therefore, soil moisture content variation within soil bodies is a critical factor influencing failure

mechanics, causing landslides, shallow slide failures, and surface erosion [1–6]. Central Mississippi's highway slopes and embankments are subject to increased risks from soil moisture variation due to the presence of expansive clay [7,8]. Rainfall and temperature patterns induce wet–dry cycles that cause and propagate cracks [2,9] and cause high volume changes in expansive clay. Near-surface soil moisture content variation and matric suction are significant factors in the frequent shallow slope failures observed on Mississippi's highway slopes [10].

Highway embankment fill slopes in Mississippi are often built with expansive Yazoo clay, which can be easily susceptible to shallow and deep-seated slide failures, especially during increased rainfall [2,7–9]. Mississippi Department of Transportation's state studies report over forty annual side slope (embankment) failures. Rainfall can increase the soil moisture content, lowering the matric suction and increasing the pore water pressure, which reduces the soil's shear strength and friction angle, making it more susceptible to failure.

As a result, constant monitoring of soil moisture content is essential for a successful geotechnical asset management program. Soil moisture content (gravimetric or "w"; volumetric or "θ") monitoring is traditionally done through single-point or precise location measurements [11]. The gravimetric method of soil moisture measurements is widely accepted for accurate measurements of "w". However, the gravimetric method is labor-intensive and destructive.

Capacitance probes, soil moisture in situ sensors such as Time Domain Reflectometry (TDR), and Frequency Domain Reflectometry (FDR) are other contact-based methods for measuring θ. Tensiometers, FDR, and TDR have largely been implemented in agricultural and grassland management applications to monitor moisture content variations [12,13]. Despite producing good results, these methods are quite expensive [14,15]. In situ point monitoring techniques such as moisture sensors lack spatial resolution and fail to give a complete picture of soil moisture variation across the slopes and embankments [14]. Installation of the sensors is expensive; moreover, the sensors come with a life span and are susceptible to premature failure due to wear and tear.

In contrast, Unmanned/Uncrewed Aerial Vehicles (UAVs) have higher spatial and temporal resolutions, enabling more efficient monitoring of site conditions [14,16]. UAVs are increasingly used in low-altitude remote sensing applications and offer many benefits such as high spatial and temporal resolution and high-quality georeferenced data [17,18]. Several studies have used UAV images to estimate soil moisture content [17,19,20]. Visible color spectra of digital images have been used to predict soil moisture content [14,20–23].

UAV optical images combined with machine learning methods have also been successfully implemented to predict soil moisture content. For instance, soil moisture prediction models have been developed using RGB images and artificial neural networks [12,24]. Machine learning techniques implemented for soil moisture prediction from remote sensing data include Multiple Linear Regression (MLR) [14], Support Vector Regression (SVR) [22,25,26], and Extreme Gradient Boosting (XGB) [27–29]. High-resolution UAV images in tandem with XGB have been applied to infer soil moisture content in precision agriculture applications [30].

Hajjar et al. [22] evaluated MLR and SVR models to predict moisture levels based on RGB color model pixels extracted from soil digital images and known moisture levels at a vineyard in Lebanon. He et al. [27] and Tao et al. [29] evaluated Random Forest and XGB models for predicting soil moisture using inputs from earth observatory satellite systems' imagery. In both studies [27,29], prediction results were evaluated in terms of performance metrics such as Root Mean Square Error (RMSE) and correlation coefficient, and XGB provided better results than other models. Kim et al. [14] developed a soil moisture content estimation equation based on the RGB color of the soil.

Alternatively, thermal sensing is another proven method to estimate soil moisture content. The underlying concept here is that the thermal inertia of the soil is negatively correlated with the variation of θ [31]. Land surface temperatures obtained from thermal images can be used for predicting soil moisture [31–33].

Inferencing soil moisture using remote sensing data and machine learning algorithms has been implemented in agriculture [16,20,22,23,30], hydrology [12,22,25], and grassland management applications [17,24].

However, using UAV-based images to characterize the soil moisture content of highway slopes and embankments is rare and needs further exploration. The soil moisture content prediction models implemented by other studies are specific to the local soil types, biological and chemical conditions, and local climate [14,22,23]. The lack of similar studies for Mississippi's soils is an added motivation to undertake this study focusing on soil moisture estimation for Mississippi's highway embankments and slopes.

The current study's objective was to develop two separate methods to predict soil moisture content ('θ' or 'w') using UAV optical and thermal images combined with machine learning and statistical modeling. The first method used RGB color values from UAV-captured optical images for predicting θ. Support Vector Machine for Regression (SVR), Extreme Gradient Boosting (XGB), and Multiple Linear Regression (MLR) models were trained and evaluated for predicting θ from RGB values. The second method used Diurnal Land Surface Temperature (ΔLST) variation from UAV-captured Thermal Infrared (TIR) images to predict θ. The two methods (RGB vs. θ and ΔLST vs. θ) were developed independently, and integrating the two methods was not within the scope of this study. This investigation used a DJI-Matrice 200 V2 drone with FLIR's Zenmuse-XT2 dual sensors to capture optical and thermal infrared images of highway embankments in Jackson, MS, USA.

## 2. Materials and Methods

### 2.1. Highway Embankment Site Locations

Highway embankment sites were chosen within a 25-mile radius of Jackson, Mississippi (Figure 1). The slopes of Sowell Road, Terry Road, and Metro Center were selected as representative highway embankments for the region. Multiple surveys using UAVs were carried out at these sites.

### 2.2. Study Design

The schematic of the study is presented in Figure 2. Two different methods were implemented to develop soil moisture prediction models. The first method used raw RGB color values from optical images to train Machine Learning (ML) models to infer θ. In the second method, diurnal LST differences determined from TIR were used to build regression models to estimate θ.

Deriving the soil moisture from the optical images required examining the underlying pattern between the RGB color integer values of the pixels and ground truth θ. Since the underlying pattern's rules are unclear for RGB vs. θ correlations, machine-learning approaches were employed to solve this problem. Specifically, SVR, XGB, and MLR methods were evaluated for θ inferencing using RGB data.

On the other hand, developing the thermal image-based θ prediction model relied on the underlying relationship between the near-surface soil thermal inertia and θ. Radiometric information embedded within the pixels of thermal infrared images was extracted using the FLIR Thermal Studio application. Then, by implementing regression modeling, the thermal inertia from infrared images captured at dawn (low temperature) and midday (peak temperature) were used to infer θ.

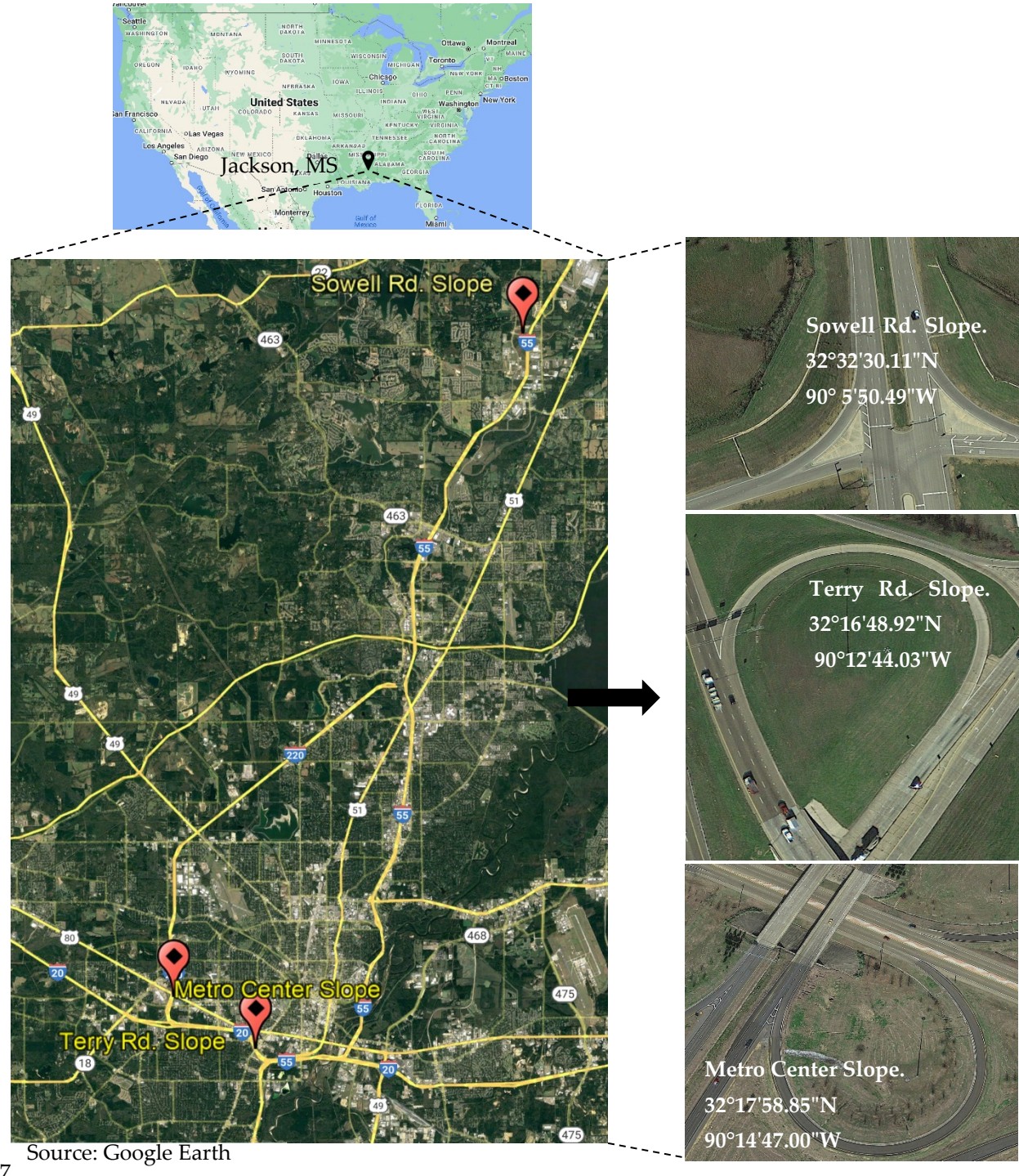

**Figure 1.** Highway embankment locations.

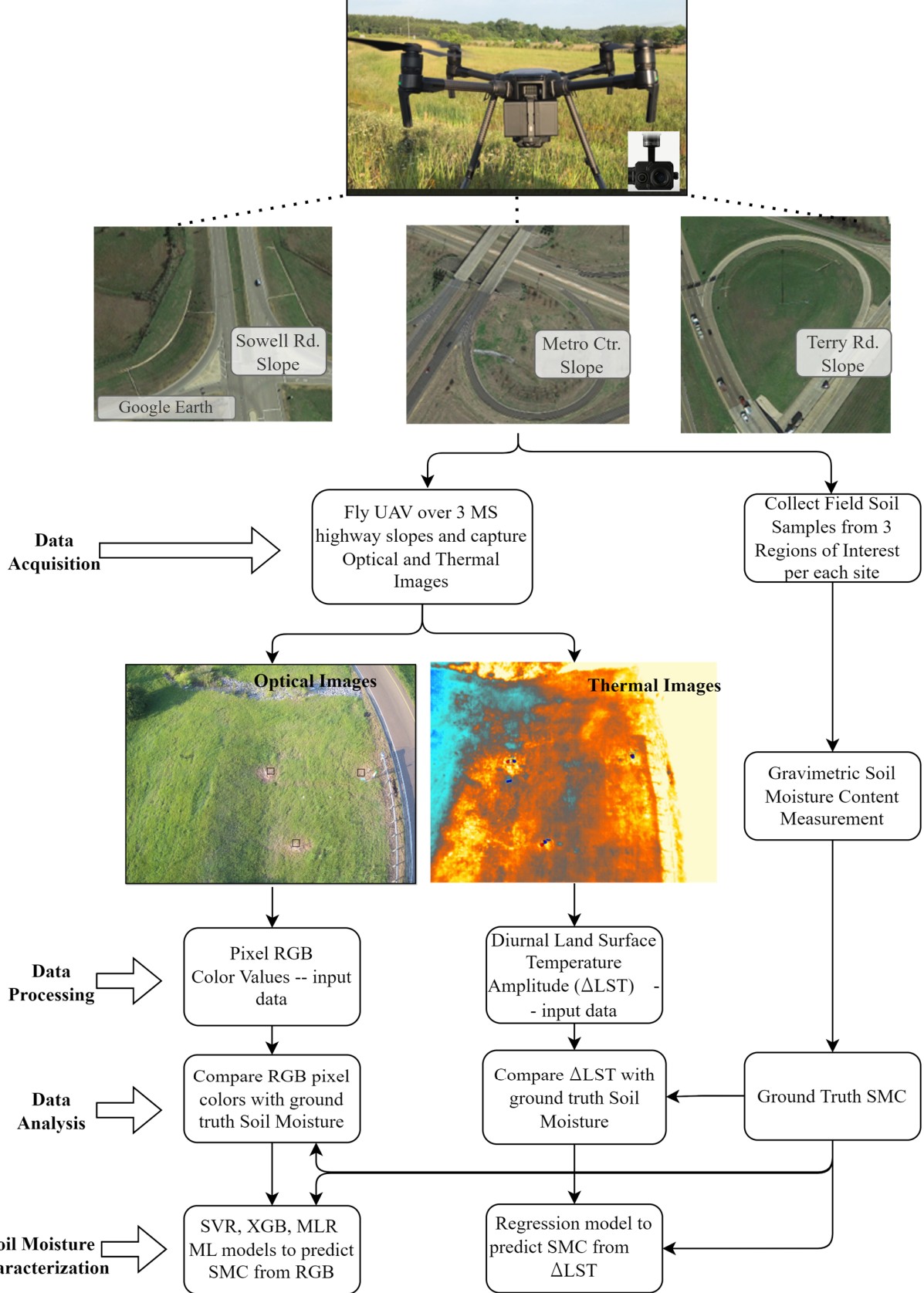

**Figure 2.** Schematic of the study.

### 2.3. UAV Data Acquisition

UAV flights were carried out by some of the authors who are FAA part 107 licensed small Unmanned Aerial Systems (sUAS) pilots who captured all the aerial optical and thermal imagery from the three sites used in this study. Images were captured using a DJI-Matrice 200 drone. The integrated 3-axis gimbal mechanism on the UAV platform helped to steady the camera during flight, reducing vibration-induced blur in the aerial photos. The UAV was flown at 100 to 200 ft (~30–60 m). Mobile applications recommended by the drone manufacturer were used for flight planning and control. The gimbal's pitch range was adjustable while in flight, ranging from −90° (i.e., nadir) to +30°.

A Zenmuse-XT2 Dual sensor camera with an optical camera and an uncooled microbolometer FLIR thermal sensor was used. The FLIR thermal sensor captures surface radiance energy by passing and collecting the Long Wave Infrared (LWIR) band of the electromagnetic spectrum from the land. Optical and thermal images were acquired on several occasions at three sites over two months in the summer of 2022. Thermal images were captured at midday and dawn to record the diurnal land temperature difference.

### 2.4. Ground Truth Soil Moisture Content

Near-surface soil samples at 1″ to 6″ (~2.5 to ~15 cm) depth were collected from three Regions of Interest (ROI) at each of the three sites, making a total of nine ROIs. These ROIs, designated as I1, I2, and I3 at each site, are presented in Figures 3–5.

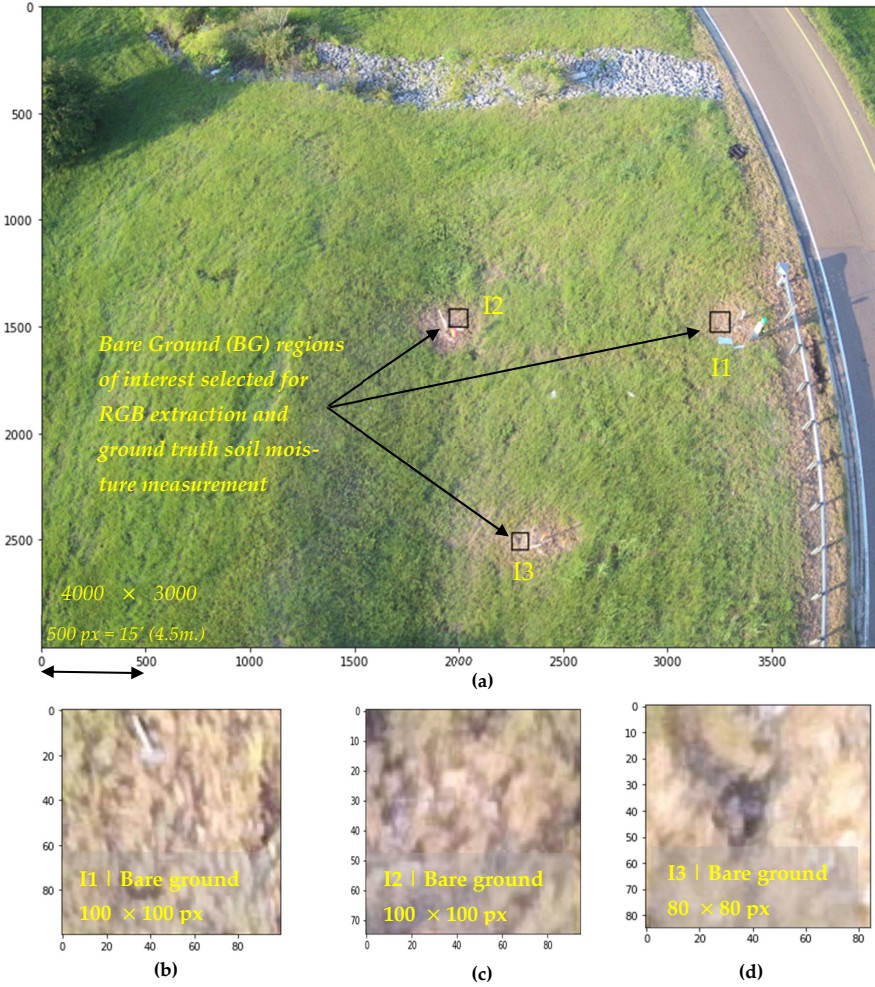

**Figure 3.** (**a**) Metro Center Highway slope UAV optical image. (**b**) Region of interest 1. (**c**) Region of interest 2. (**d**) Region of interest 3.

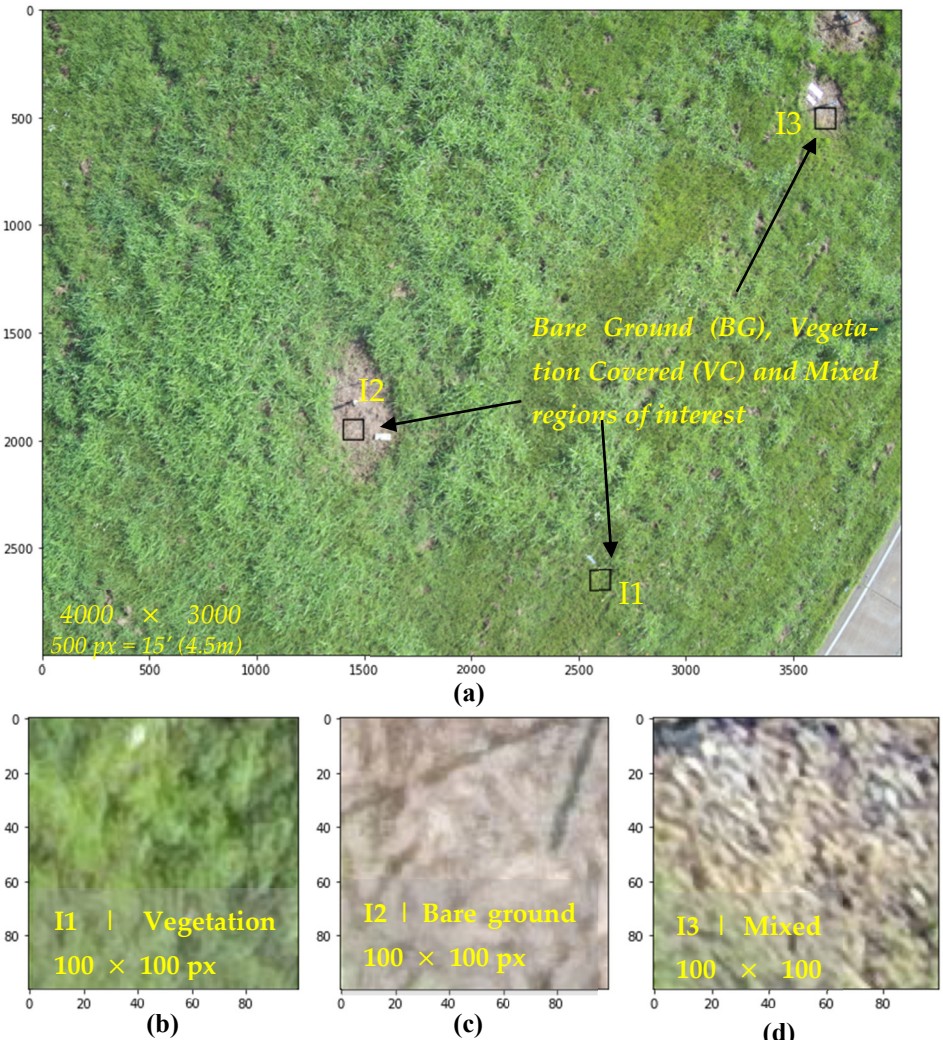

**Figure 4.** (**a**) Terry Road slope UAV optical image. (**b**) Region of interest 1. (**c**) Region of interest 2. (**d**) Region of interest 3.

ROIs selected included Bare Ground (BG), Vegetation Covered (VC), and mixed BG plus VC locations. Such multiclass ROIs were purposely selected to help generalize the θ inferencing capabilities of the developed models across both types of image classes. The average gravimetric soil moisture content (*w*) for each ROI was determined in the laboratory per ASTM (2010) Standard D2216 as described in the following steps. Step 1: weigh a labeled empty container ($w_c$) with the lid on. Step 2: add the collected moist soil into the container, close the lid, and measure weight ($w_1$). Step 3: remove the lid and place the container with the soil sample in the oven for up to 24 h at a temperature of 230° F (~110° C). Step 4: remove the container from the oven, close the lid, and measure the weight ($w_2$). Steps 1 through 4 were performed for soil samples collected from nine ROIs in total from all sites. The ground truth *w*% was determined by Equation (1).

$$w\% = \frac{W_w}{W_s} \times 100\% \tag{1}$$

where *w* = gravimetric soil moisture content, ($w_w$ = weight of water determined by ($w_1 - w_2$), and $w_s$ = weight of solids determined by ($w_2 - w_c$),

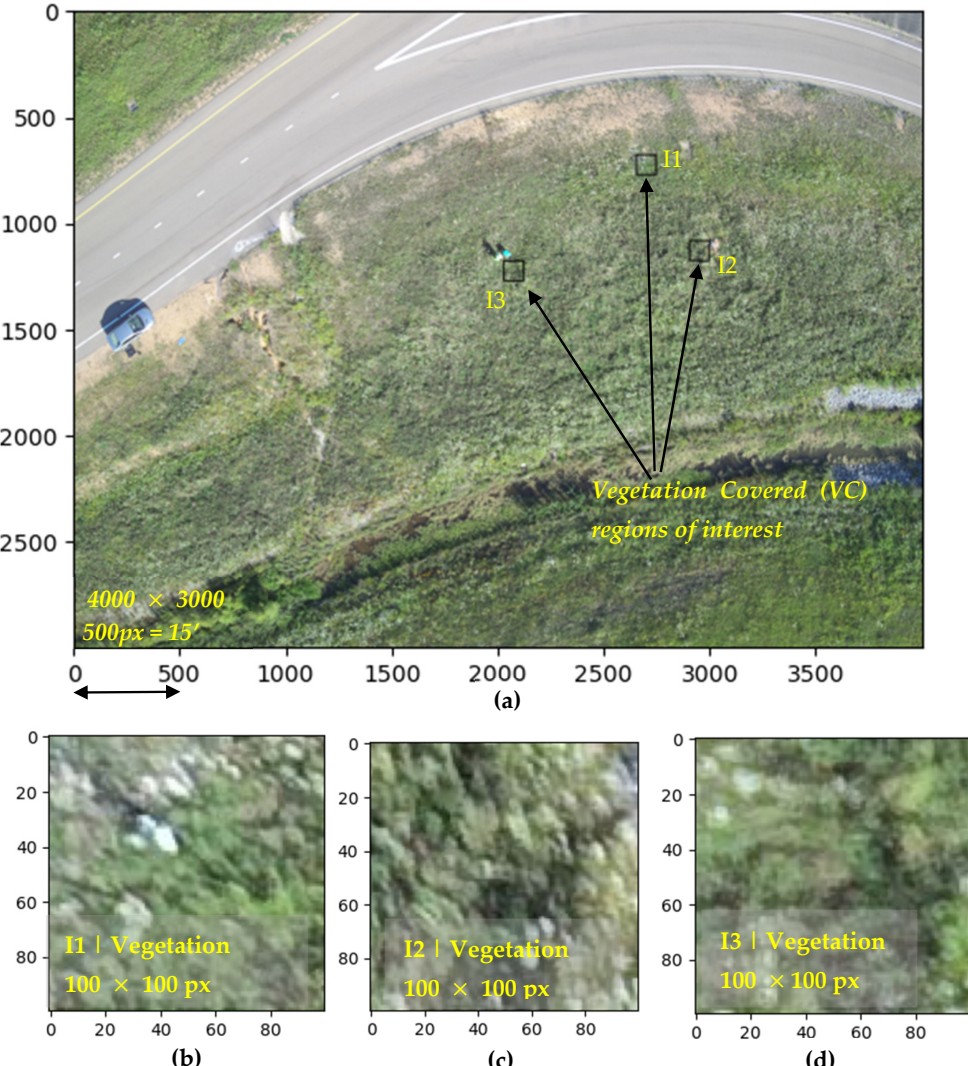

**Figure 5.** (**a**) Sowell Road slope UAV optical image. (**b**) Region of interest 1. (**c**) Region of interest 2. (**d**) Region of interest 3.

The ground truth volumetric moisture content (θ) was determined using Equation (2).

$$\theta = \frac{SwG_s}{(S + wG_s)} \tag{2}$$

where θ = volumetric soil moisture content, $w$ is gravimetric soil moisture content, $G_s$ is the specific gravity of 2.7, and the degree of saturation S of 50% was considered based on a past study at these sites [7].

Ground truth $w$ and θ determined at each site at the selected ROIs are presented in Table 1.

**Table 1.** Ground truth soil moisture content.

| Slope | ROI | Class | $w$% | $\theta$% |
|---|---|---|---|---|
| Metro | 1 | Bare ground (BG) | 20% | 26% |
| Metro | 2 | Bare ground (BG) | 19% | 25% |
| Metro | 3 | Bare ground (BG) | 11% | 18% |
| Terry | 1 | Vegetation covered (VC) | 14% | 21% |
| Terry | 2 | Bare ground (BG) | 23% | 28% |
| Terry | 3 | Mixed | 16% | 23% |
| Sowell | 1 | Vegetation covered (VC) | 14% | 22% |
| Sowell | 2 | Vegetation covered (VC) | 12% | 20% |
| Sowell | 3 | Vegetation covered | 13% | 21% |

## 2.5. UAV Optical Image Data Processing and Analysis

The UAV-captured optical images were examined, and appropriate aerial images without any blur, shadows, and distortions were selected for RGB extraction. The optical image resolution was 3000 × 4000 pixels (px), and the ground resolution of the optical images was approximately 0.03 ft. (0.009 m)/px. Three ROIs per site ranging from 80 × 80 px ~100 × 100 px areas were identified, making a total of nine ROIs from three sites. These ROIs were at the same locations from where soil samples were collected to determine ground truth soil moisture content. The ROIs selected included some bare-ground areas as well as vegetation-covered areas.

The Metro Center slope image and the three bare-ground ROIs are presented in Figure 3. The Terry Road slope image with the multiclass ROIs (one bare ground, one Vegetation covered, and one mixed class) are presented in Figure 4. Such multiclass ROIs were selected to help generalize the $\theta$ inferencing capabilities of the developed models across different types of image classes. The Sowell Road slope image with the three vegetation-covered ROIs are presented in Figure 5.

### 2.5.1. RGB Extraction and Visualization

The raw RGB color integer values embedded in each pixel of the optical images are a combination of integer values that range from 0 to 255. RGB raw values from every 10th pixel in horizontal and vertical directions were extracted from the 100 × 100 px ROIs. From the 9 ROIs (3 per site), a total of 879 rows of RGB data vectors were extracted. Out of the 879 data vectors, 381 belonged to the Bare Ground (BG) class, 400 to the Vegetation Covered (VC) class, and 98 to the mixed class.

The raw RGB values extracted from the ROI were compared with ground truth $\theta$ for the corresponding ROI. Figure 6a presents the variation of all 879 RGB data vectors vs. the average $\theta$ extracted from the 9 ROIs off the 3 sites. After averaging, the multiclass RGB data vectors boil down to nine instances, with four belonging to the bare ground class, four to the vegetation covered class, and one to a combined class. The variation of RGB averages with average soil moisture content at their respective ROIs is presented in Figure 6b. The average RGBs at each ROI and corresponding ground truth soil moisture content values are also presented in Table 2.

**Table 2.** Average RGB and soil moisture content variation at 9 ROI.

| Site | ROI | Class | GSMC ($w$) | VSMC ($\theta$) | R | G | B |
|---|---|---|---|---|---|---|---|
| Metro | 1 | BG | 20% | 26% | 203.59 | 179.53 | 155.27 |
| Metro | 2 | BG | 19% | 25% | 183.88 | 168.81 | 141.26 |
| Metro | 3 | BG | 11% | 18% | 199.78 | 182.25 | 152.32 |
| Terry | 1 | VC | 14% | 21% | 122.93 | 150.44 | 86.32 |
| Terry | 2 | BG | 23% | 28% | 188.49 | 176.58 | 165.23 |
| Terry | 3 | BG + VC | 16% | 23% | 174.75 | 169.77 | 150.75 |
| Sowell | 1 | VC | 14% | 22% | 130.40 | 148.35 | 118.89 |
| Sowell | 2 | VC | 12% | 20% | 121.55 | 132.60 | 106.98 |
| Sowell | 3 | VC | 13% | 21% | 113.18 | 128.12 | 97.65 |

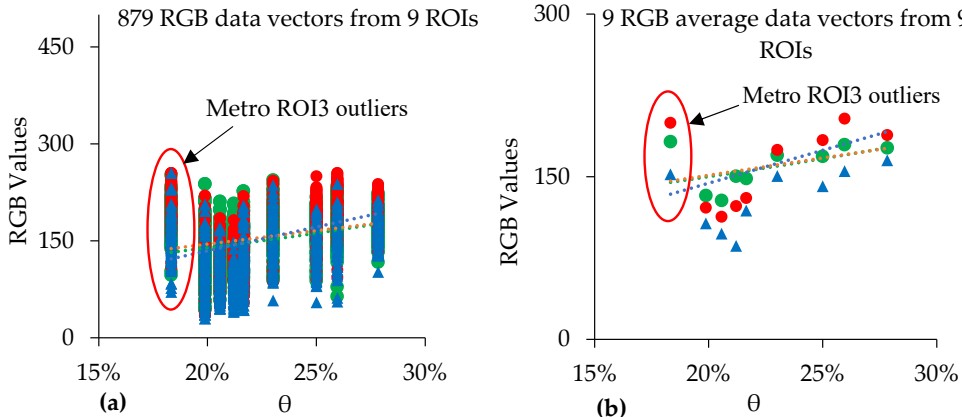

**Figure 6.** (**a**) RGB data vectors (all sites) variation with θ (**b**) RGB averages (all sites) variation with θ.

The RGB average values variations with θ separated by the BG and VC classes are presented in Figure 7. Figure 7a represents the average RGB vs. θ for bare-ground ROIs, and Figure 7b represents average RGB values vs. θ for vegetation-covered ROIs. The red color is higher for the bare ground class, and the green color is higher for the vegetation-covered images, which is plausible. Furthermore, the near-surface soil moisture content was observed lower in vegetation-covered slope areas. In contrast, the near-surface θ was observed higher on bare-ground areas of the slope. This observation indicates that vegetation captures the soil moisture from the near-surface soil, resulting in lower near-surface.

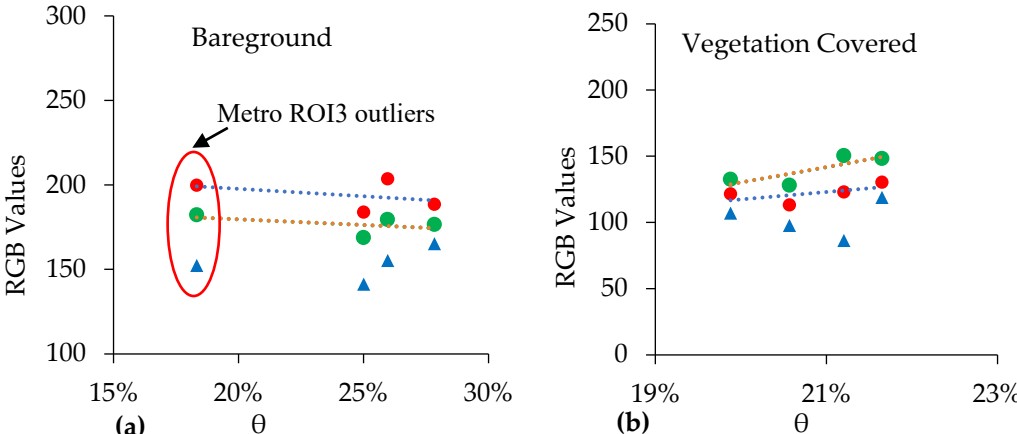

**Figure 7.** RGB$_{avg}$ variation with θ for all sites. (**a**) Bare ground class pixels. (**b**) Vegetation covered class.

### 2.5.2. Data Cleaning

As the moisture content increases, the soil color should also increase [21] (Persson, 2005). However, for the Bare Ground (BG) class in Figure 7a, the RGB variation does not follow this logic and has higher color values for low soil moisture content, mainly due to the outliers. Similarly, it is also evident from Figure 6b that the RGB values obtained from metro center ROI#3 behaved as outliers and therefore were removed from the RGB dataset.

Furthermore, the RGB dataset was further examined, and more outliers were identified to avoid errors in the soil moisture content estimating models. The following procedure was implemented to remove the outliers. Firstly, the sums of the R, G, and B values were calculated for each of the 879 rows. Then, the sums of the RGB values were divided into 4 quartiles, and 25th, 50th, and 75th percentiles were calculated. The RGB sum values lesser than the 25th percentile and greater than the 75th percentile were considered outliers and eliminated from the dataset. The mixed class data from Terry Road ROI#3 were also left out from the cleaned-up dataset. After removing the outliers, 370 rows of RGB data vectors remained, consisting of 169 BG class and 201 VC class. All subsequent results were derived

using the cleaned-up dataset without outliers. The 370 rows of RGB, together with the ground truth soil moisture content, were used to train the machine learning models. The 370 RGB data vector distribution is presented in Figure 8.

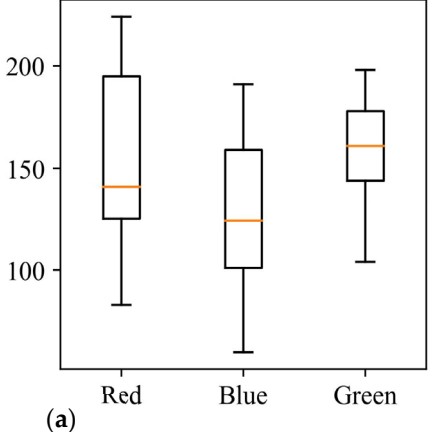
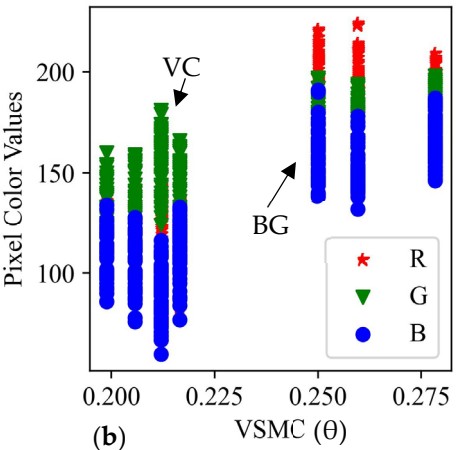

**Figure 8.** Cleaned-up dataset visualization. (**a**) RGB boxplots. (**b**) Vegetation vs. bare ground RGB data distribution.

### 2.5.3. Machine Learning Methods for Soil Moisture Content Characterization Using RGB Data

A total of 879 rows of RGB data were extracted from the optical image ROIs for the 2 classes of bare ground and vegetation covered. The final cleaned-up dataset was about 370 rows of RGB values and corresponding ground truth moisture content values. The ML models SVR, XGB, and MLR were developed using the cleaned-up dataset to infer soil moisture content. Each ML method was trained in different ways in terms of RGB data classes: one with combined classes of Bare Ground (BG)and Vegetation Covered (VC), the second with the bare ground class alone, and the third with the vegetation covered class alone. Jupyter Notebook and the Python programming language were used to perform data cleaning and visualization and to develop the machine learning models.

Training-Test Split: The cleaned-up dataset was split into 80% training and 20% test samples using the train_test_split function imported from the sklearn library to develop the machine learning models. For the VC+BG combined class dataset, out of the 370 data vectors, 296 RGB data vectors were used for training and 74 data vectors were used for testing. For the BG-only dataset, out of the 169 data vectors, 135 RGB were used for training and 34 were used for testing. For the VC-only dataset, out of the 201 data vectors, 160 RGB data vectors were used for training and 41 data vectors were used for testing.

Support Vector Regression (SVR): A Support Vector Machine (SVM) is a proven machine learning algorithm for classification and regression. When used for regression, it is known as SVR. SVR has been used to develop soil moisture prediction models based on RGB values for lab-scale models [22] (Saad Hajjar et al., 2020) and for the region scale using satellite imagery [12] (Ahmad et al., 2010). In this study, e-SVR was implemented to predict soil moisture from the RGB input features. The support vector regression in its standard form is presented in Figure 9. SVR allows the user to decide how much error in the model is acceptable, and it will locate a suitable line in two-dimensional problems (or hyperplane in higher dimensions) to fit the data. In Figure 9, epsilon (e) represents the allowed error margin. Epsilon can also be visualized as the tube that determines the hyperplane width.

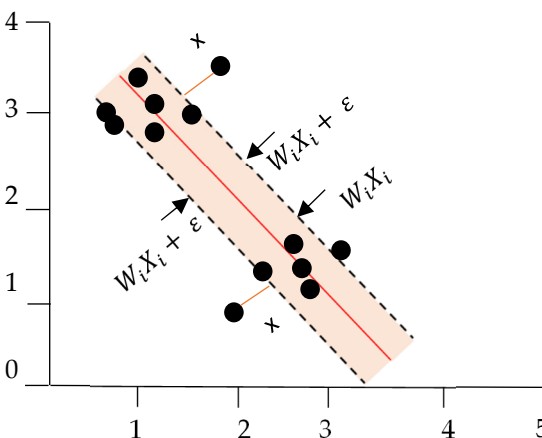

**Figure 9.** Support vector regression (typical form).

The goal would be to predict data within the given margin of error (e). Variables beyond the margin lines are assigned a deviation value called slack ($\xi$). The other significant parameters include kernel, regularization factor C, and gamma. RBF kernel function was used in this study, which allows implementing regression using hyperplane on a higher dimension. The most widely used regularization factor is C = 1.0 [34]. The higher the C, the better the fitting on the training dataset. However, a higher C impedes the model's generalization ability and affects the test dataset predictions.

*Extreme Gradient Boost (XGBoost or XGB) method:* XGB is a tree-based ensemble machine learning algorithm that improves on the gradient boosting framework by incorporating certain precise approximation algorithms [35]. It offers improved prediction power and performance. Ensemble machine learning modeling approaches such as XGB are powerful models made up of a group of weaker base models [36]. It uses gradient boosting to construct machine learning algorithms. It is an advanced implementation of the Gradient-Boosted Regression Tree (GBRT) [37], which is a sequential ensemble method that adds several base regression trees over time to increase the capacity of the entire model. This ensemble approach can be used for regression. High-resolution UAV images in tandem with XGBoost algorithm have been applied to infer soil moisture content [30]. Among the many parameters within XGBoost architecture, the following make a difference to the prediction results:

- Gamma: loss function varying between 0 and 1.
- Max Depth [0~infinity]: Maximum depth of a tree. A default value of 6 is set but varied up to 100 and observed for changes. Increasing this value will make the model more complex and more likely to overfit.
- Min Child Weight [0~infinity]: for linear regression, this refers to a minimum number of instances that are needed in each node.
- Learning Rate [0~infinity]: the model's learning rate determines how quickly it adapts to the given problem and is, by default, 0.3.

*Multiple linear regression method:* Multiple linear regression models are an excellent choice for simple problems for quantifying relationships between a set of independent variables (features) and a single dependent variable (target). Therefore, MLR models were implemented for soil moisture prediction using RGB values in this study. A typical MLR model is shown in Equation (3).

$$y = \sum_{0}^{n} a_n x_n \tag{3}$$

where y is the target and $x_0 = 1$, $\{x_1 \sim x_n\}$ are input features, $a_0$ is the slope intercept, and $\{a_1 \sim a_n\}$ are the coefficients. Similar to the other machine learning models, three multiple linear regression models were developed, one for the combined class model (MLR1), one for the bare ground class (MLR2), and one for the vegetation class (MLR3).

### 2.6. UAV Thermal Infrared (TIR) Image Processing

The thermal sensor payload captured TIR images with a $640 \times 512$ px resolution. The TIR images were processed through the FLIR Thermal Studio application to obtain pixel temperature values. The temperature values were then corrected (commonly referred to as calibration) for the ambient conditions. The radiometric data were calibrated for the observed humidity, ambient temperature, reflected temperature, and emissivity.

#### 2.6.1. TIR Temperature Calibration, Verification, and Optimization

TIR Calibration: Thermal sensor calibration is a prerequisite for accurate radiance and absolute temperature measurements for thermography applications. The FLIR Zenmuse XT2 (FZ2) thermal sensor used in this study is an uncooled microbolometer type, and the manufacturer calibrates it during production to carry out infrared radiance and temperature measurements. FZ2 has been drift compensated by the manufacturer, which means that the camera compensates the output for variation in the camera's internal temperature. FZ2 has also gone through verification against a standard traceable black body. This verification process aims to compare the radiance from the TIR against a known traceable blackbody ground truth, validate the temperature measurements, and adjust the sensor accordingly. The TIR images captured by FZ2 were post-processed in the FLIR Thermal Studio software application. During post-processing, the absolute land surface temperature from raw thermal images is read and converted to infrared radiance emitted from surfaces using Planck's Equation.

We further verified the thermal images and adjusted them to specific field conditions by adjusting the object parameters: ambient temperature, relative humidity, emissivity, and reflected temperature. For the surficial soil, an emissivity of 0.8–0.95 was used based on published data from FLIR. Relative humidity and ambient temperature values were obtained from the data published by NOAA for the nearest weather station for the time of the day the UAV flight occurred. The weather station parameters were input into the FLIR Thermal Studio application, which has inbuilt algorithms that correct the LST based on the object parameter values. The reflected temperature adjustment was performed by following the standard reflector test method explained in the following paragraph.

Reflected temperature is any thermal radiation originating from other objects that reflects off the target measured. The value of the reflected temperature should be calculated and programmed into the camera's parameters to make it possible for the software to compensate and ignore the effects of this radiation to obtain the actual surface temperature of the soil. The reflected temperature is related directly to the emissivity of the same object; higher emissivity objects tend to produce less reflected temperature influence. Therefore, objects with a lower emissivity (such as aluminum) and high reflectance can provide an accurate measure of the atmospheric reflected temperature.

ASTM E1862 describes the commonly adapted reflector method using aluminum foil (1 ft wide $\times$ 2′ long) for assessing the reflected infrared energy from the atmosphere [38]. This method has been successfully implemented in previous relevant studies by [39].

In the current study, aluminum (AL) foil was placed on the field during the UAV flight, as shown in Figure 10. The UAV-mounted thermal sensor FZ2 captured the temperature of the aluminum foil reflector. In post-processing, the emissivity value of the AL foil is set to 1, and the distance is set to 0. Then, the average surface temperature reading of the AL foil reflector is measured. This surface temperature value of the AL foil reflector is assigned to its reflected temperature, and the foil's surface temperature measurement is retaken. The resulting temperature provides the actual atmosphere-reflected temperature for the specific site on that specific day. All thermal images captured on that site at a specific time were adjusted according to the atmospheric reflected temperature obtained from the standard reflector test procedure. Aluminum foils and AL markers also help to locate regions of interest on the thermal images during post-processing.

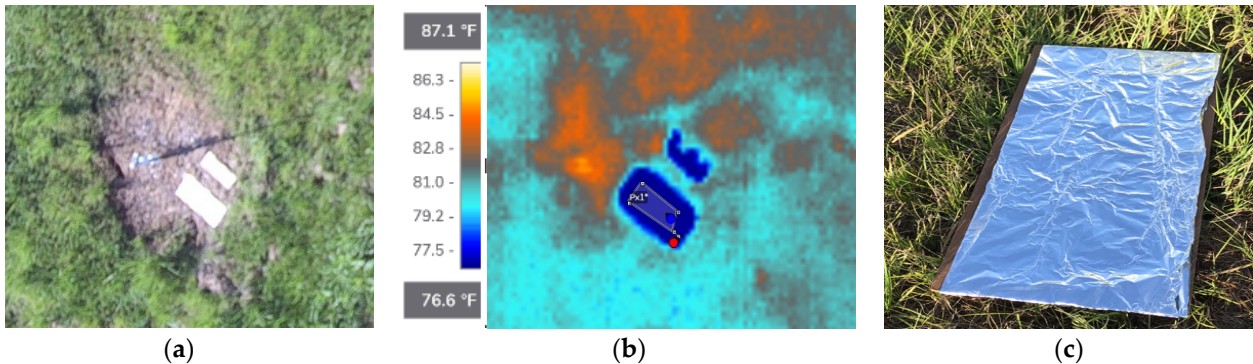

(**a**)　　　　　　　　　　　　　　(**b**)　　　　　　　　　　　　　　(**c**)

**Figure 10.** (**a**) Optical image of AL foil on the Terry Road site. (**b**) AL foil thermal image of the Terry Road site. (**c**) Typical AL foil placed on site.

The post-calibrated and optimized object parameters finalized in FLIR Thermal Studio are presented in Table 3. Once these object parameters were finalized and inputted into the FLIR Thermal Studio tool, the LST for the ROI locations were readily obtained. Then, the LSTs were used to calculate the diurnal LST amplitudes.

**Table 3.** TIR image object parameters.

| Parameter | Metro | | Terry | | Sowell | |
|---|---|---|---|---|---|---|
| | Dawn | Midday | Dawn | Midday | Dawn | Midday |
| Distance (ft.) | 100 | 100 | 100 | 100 | 100 | 100 |
| Angle of view (deg.) | 60 | 60 | 90 | 90 | 60 | 60 |
| Relative humidity | 82% | 48% | 82% | 48% | 93 | 46 |
| Ambient temperature (F) | 79 | 94 | 79 | 94 | 72 | 87 |
| Emissivity of the scene | 0.92 | 0.92 | 0.92 | 0.92 | 0.9 | 0.92 |
| Reflected temp. AL foil (F) | 38.7 | 52.5 | 48.9 | 73.8 | 26.8 | 11.3 |
| Precipitation (in.) | 0 | 0 | 0 | 0 | | 0 |

### 2.6.2. Diurnal LST Data Modeling for Soil Moisture Content Characterization

The diurnal LST difference is calculated by taking the temperature difference between peak temperature and low temperatures within the day. Diurnal LST difference is inversely related to the soil moisture content [31]. This underlying relationship was used to develop a regression model to estimate $\theta$.

LST verification and diurnal temp range: Maximum LST was taken during midday peak temperatures. The temperature verification passed, as the TIR measured matched the soil probe temperature. LST min at dawn did not agree with the probe after the initial object parameter setup. The high relative humidity causes radiation scattering and impacts sensors' accuracy in reading the radiation. To accommodate this scatter, TIR object parameters were adjusted. An emission of 0.82 brought LST closer to the probe temperature. Therefore, the emissivity of 0.82 was used for all images captured at dawn. LST was validated by comparing the pixel temperatures with the soil surface temperature measured using a probe thermometer. The comparison provided a good match, with a low Root Mean Square Error (RMSE) of 2.5, as shown in Figure 11.

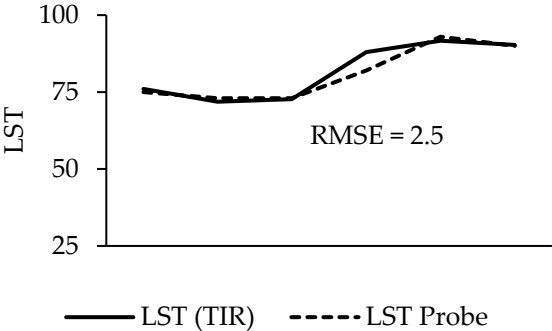

**Figure 11.** LST validation.

The typical dawn and midday thermal images next to their optical counterpart are presented in Figure 12. The thermal images were optimized for better visualization by selecting the FLIR Thermal Studio's inbuilt arctic color palette and temperature linear color distribution during post-processing.

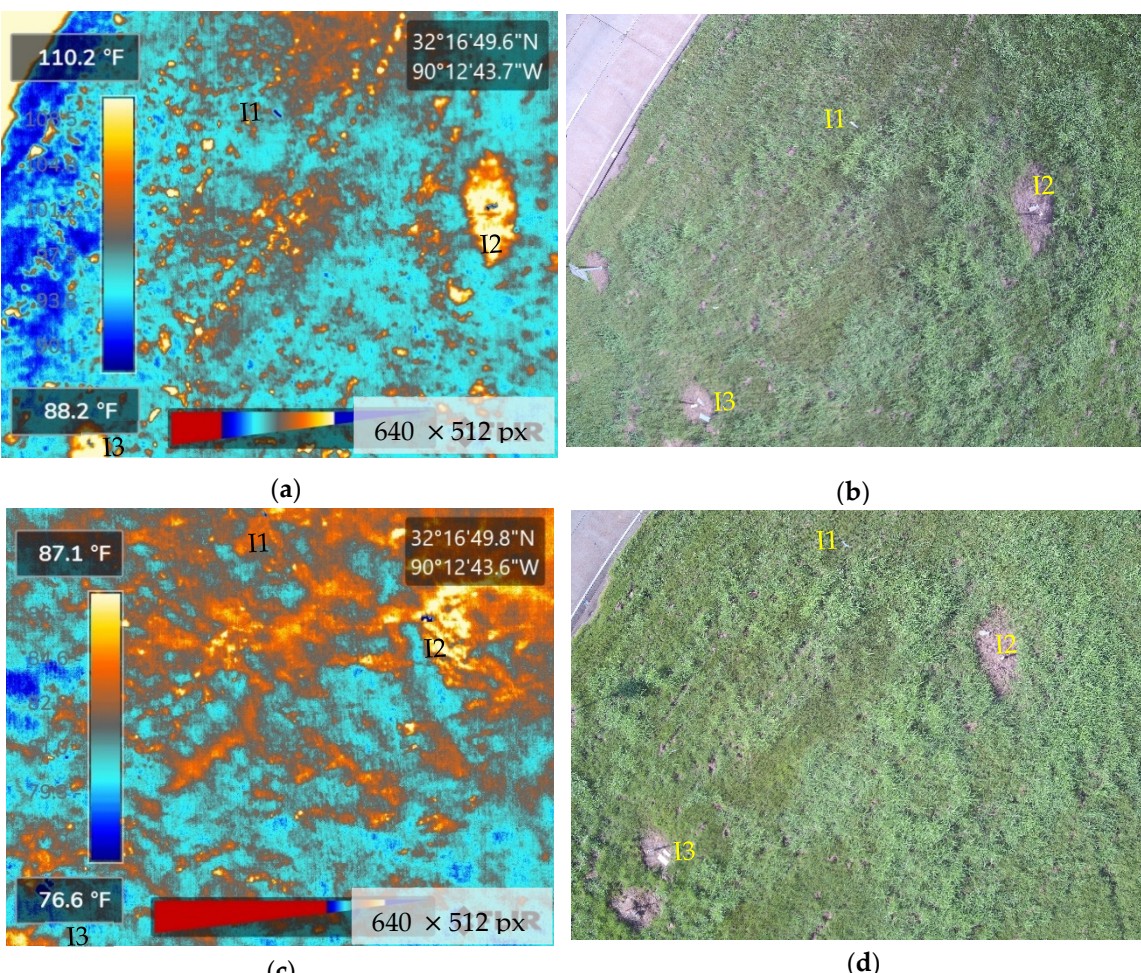

**Figure 12.** Thermal image and optical images at Terry Road. (**a**,**b**) Dawn. (**c**,**d**) Midday.

### 3. Results

*3.1. Soil Moisture Characterization Using RGB Data from Optical Imagery*

3.1.1. SVR Results

Support vector regression models were trained and developed to predict soil moisture from RGB color values from the optical images. As noted above, the study considered two classes of images, one for Bare Ground (BG) and one for Vegetation Covered (VC). Therefore, three models were built. SVR1 was developed with a combined dataset consisting of images of both vegetation and bare ground classes. SVR2 was built for bare ground class images, and SVR3 for vegetation class.

The values of the model parameters were adjusted in several iterations, and the performance of the model was recorded. The models were evaluated based on the R2 score for the training and test datasets. The metrics used to determine the model performance were the R2 score, the mean absolute error, the mean square error, and the root mean square errors. Finally, the set of parameters or the combination of parameters that provide the best results in terms of the R2 score was selected for the model. The results of various iterations of SVR model parameters and the resulting prediction scores (R2) are presented in Table 4.

**Table 4.** SVR model parameters and performance metrics *.

| | Class | Kernel | C | Epsilon | Gamma | R2-trn | R2-test | Test MSE | Test RMSE | Test EVS |
|---|---|---|---|---|---|---|---|---|---|---|
| SVR1 | VC + BG | rbf | 1 | 0.01 | auto | 0.88 | 0.25 | $1 \times 10^{-9}$ | $2 \times 10^{-7}$ | $2 \times 10^{-5}$ |
| SVR2 | BG | rbf | 1 | 0 | auto | 0.99 | 0.25 | $6 \times 10^{-8}$ | $5 \times 10^{-7}$ | $9 \times 10^{-4}$ |
| SVR3 | VC | rbf | 1 | 0 | auto | 0.99 | 0.03 | $4 \times 10^{-5}$ | $1 \times 10^{-7}$ | $4 \times 10^{-5}$ |

* Table Legend: Regression Function (Kernel), Regularization Factor (C), Allowed Error Margin (Epsilon), Loss Function (Gamma), Train Score (R2-trn), Test Score (R2-test), Mean Square Error (MSE), Root MSE (RMSE), Explained Variance Regression Score (EVS), Support Vector Regression (SVR), Vegetation Covered (VC), and Bare Ground (BG).

It is important to note that, although the SVR provided good results on the training dataset, the model failed to provide good predictions for the test dataset. Despite the good training results, the test results in terms of R2 score were poor. Furthermore, the test Explained Variance Regression Score (EVS) function was much farther from 1.0 and not indicative of healthy prediction models.

The line plot and scatter plot of measured θ versus those predicted by the SVR1 model are presented in Figure 13.

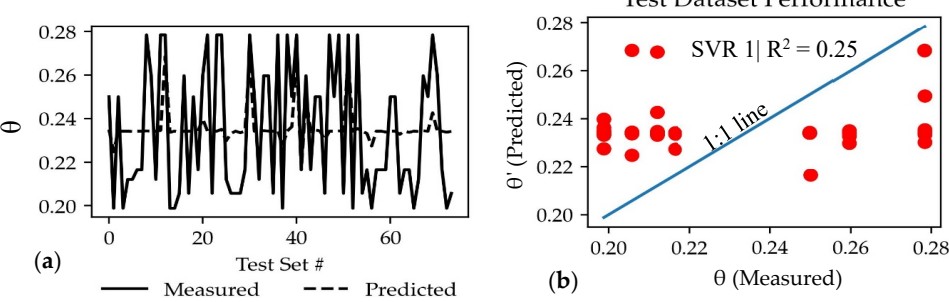

**Figure 13.** SVR1 soil moisture content evaluation metrics: measured (θ) vs. predicted (θ') for the test samples. (**a**) Line plot. (**b**) Scatter plot.

### 3.1.2. XGB Results

Three XGB regression models were developed to predict soil moisture from RGB color values from the optical images. XGB1 was developed using a combined dataset with images of both BG and VC classes. This model was trained on 296 data vectors of RGB and target θ and then tested on 74 data vectors of the RGB features set.

XGB2 was trained on 201 RGB data vectors and corresponding ground truth θ of the BG class, split into 160 for the training and 41 test sample sets. XGB3 was trained on 168 data vectors and the target θ of the VC class, which were split into 134 training and 34 test data vectors.

The following parameters were adjusted in different trial runs of the XGB models: Kernel: RBF-Gamma was varied between 0 and 1. Max Depth: 6 to 10 were explored. Min child weight [0~infinity]: between 1 and 2. The learning rate was adjusted between 0 and 0.5, but 0.3 proved to be the optimal rate.

The XGB models' performance metrics and the combination of parameters are presented in Table 5. XGB1 and XGB2 show excellent R2 scores compared to XGB3. An Explained Variance Regression Score (EVS) function of the sklearn library close to 1 represents a healthy prediction model. EVS for all three XGB models were >0.9, proving good prediction quality. Additionally, the test dataset's RMSE and MSE loss functions are close to zero, indicating well-performing models. Therefore, all three models are suitable for carrying out future predictions.

**Table 5.** XGB model parameters and performance metrics *.

| | Class | Learning Rate | Max Depth | Min Child Weight | Gamma | R2-Trn | R2-Test | Test MSE | Test RMSE | Test EVS |
|---|---|---|---|---|---|---|---|---|---|---|
| XGB1 | VC + BG | 0.3 | 20 | 2 | 0.0015 | 0.85 | 0.9 | $8 \times 10^{-5}$ | 0.0090 | 0.91 |
| XGB2 | BG | 0.3 | 10 | 2 | 0.0015 | 0.92 | 0.93 | $7 \times 10^{-5}$ | 0.0088 | 0.92 |
| XGB3 | VC | 0.3 | 6 | 2 | 0 | 0.95 | 0.4 | $2 \times 10^{-6}$ | 0.0014 | 0.95 |

* Table Legend: Learning Rate (0~infinity), Max Depth (0~infinity), Loss Function (Gamma), Train Score (R2-trn), Test Score (R2-test), Root Mean Square Error (RMSE), Explained Variance Regression Score (EVS), Extreme Gradient Boosting (XGB), Vegetation Covered (VC), and Bare Ground (BG).

The line plot and scatter plot of measured volumetric soil moisture content (θ) versus predicted (θ') by the SVR1 model are presented in Figure 14.

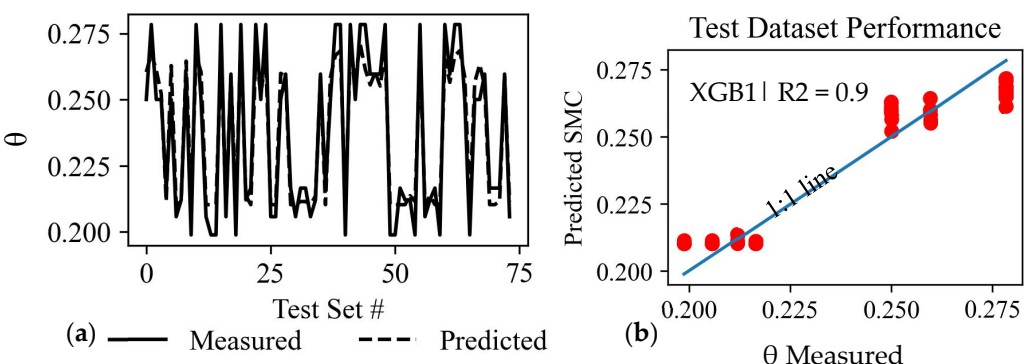

**Figure 14.** XGB1 soil moisture content evaluation metrics: measured (θ) vs. predicted (θ') for the test samples. (**a**) Line plot. (**b**) Scatter plot.

### 3.1.3. MLR Results

Three MLR models, MLR1, MLR2, and MLR3, were developed for predicting θ using RGB values of the combined class, bare ground (BG), and vegetation cover (VC) classes, respectively. As noted in Section 2.5.2, prior to data cleaning efforts, 879 RGB data vectors were initially extracted from 9 ROIs and corresponding ground through soil moisture. After data cleaning and removing the outliers, 370 rows of RGB data vectors remained. Of the 370 rows, 169 belonged to the BG and 201 belonged to the VC class. The MLR1 model was built using the cleaned-up dataset without outliers, consisting of 370 rows of combined VC + BG classes. MLR2 was built using 169 data vectors of the BG class, and MLR3 was built using 201 data vectors of the VC class. The combined VC + BG class RGB vectors and the target θ were used to construct MLR1.

The regression equation for soil moisture prediction from RGB pixel values is shown in Equation (4).

$$\theta' = a_0 + a_1 R + a_2 G + a_3 B \tag{4}$$

where $\theta'$ = predicted moisture content; R, G, and B are red, green, and blue pixel values; and $a_0$, $a_1$, $a_2$, and $a_3$ are coefficients calculated from the regression analysis.

The MLR models were developed in iterative steps. The initial MLR models were developed, including all independent variables (RGB). The model coefficients with high Standard Errors (SE) were identified and rejected, and the corresponding independent variables were eliminated from subsequent regression steps. The next round of regression was carried out with the remaining independent variables, and the resulting coefficients and the corresponding SE were re-evaluated. The iterative process was continued until favorable coefficients with low SEs were realized. The process for MLR1, for instance, is explained in the following steps:

- Step 1. After the first round of regression, the initial MLR1 model had the following coefficients and SE: $a_0$ (0.120659945, 0.006710459); $a_1$ (0.000339414, $6.37115 \times 10^{-5}$); $a_2$ ($7.52789 \times 10^{-5}$, $7.44529 \times 10^{-5}$); $a_3$ (0.000374743, $5.43763 \times 10^{-5}$); where the first parameter within the parentheses is the coefficient value and the second is the SE.
- Step 2. The standard error for $a_2$ (coefficient of Green(G)) in step 1 was as large as the coefficient value itself, so $a_2$ was rejected. As a result, the "G" independent variable column was eliminated in the next round of regression. The MLR1 model was re-built with only R and B independent variables, and the resulting coefficients and standard errors were as follows: $a_0$ (0.126713038, 0.003031591); $a_1$ (0.000383043, $4.68758 \times 10^{-5}$); $a_2$ (0,0); $a_3$ (0.000367793, $5.39417 \times 10^{-5}$).

The $a_0$, $a_1$, and $a_3$ coefficients of MLR1 after Step 2 had small SE values and thus were accepted, and the regression was stopped. However, comparing the coefficients from Steps 1 and 2, it is apparent that they remained almost identical even though MLR1 was rebuilt after eliminating the "G" independent variables.

The regression statistics of the overall MLR1 models developed in the two steps are presented in Table 6. The multiple or "Pearson" correlation coefficient (R) for both iterations of MLR1 was 0.89, indicating a positive correlation. The adjusted R square value was 0.79, indicating that the independent variables (mainly R and B) accounted for 79% of the variance in soil moisture prediction. Therefore, a high effect is reported by the model. The Mean Square Error (MSE) and Root Mean Square Error (RMSE) for MLR1 improved only slightly after rejecting $a_2$, thus confirming that the "G" independent variable had a minuscule impact on $\theta'$ prediction of the MLR1 model. The fact that the weight of the G variable was so small ($a_2 = 7.5 \times 10^{-5}$) explains its minimal effect on model output. The distribution of residuals (predicted—measured) for MLR1 is presented in Figure 15.

**Table 6.** MLR1 regression statistics and performance metrics.

|  | MLR1 (Step 1) | MLR1 (Step 2) |
|---|---|---|
| Multiple Correlation Coefficient Value (R) | 0.89 | 0.89 |
| R Square | 0.79 | 0.79 |
| Adjusted R Square | 0.79 | 0.79 |
| MSE | 0.00016 | 0.00007 |
| RMSE | 0.012 | 0.01 |

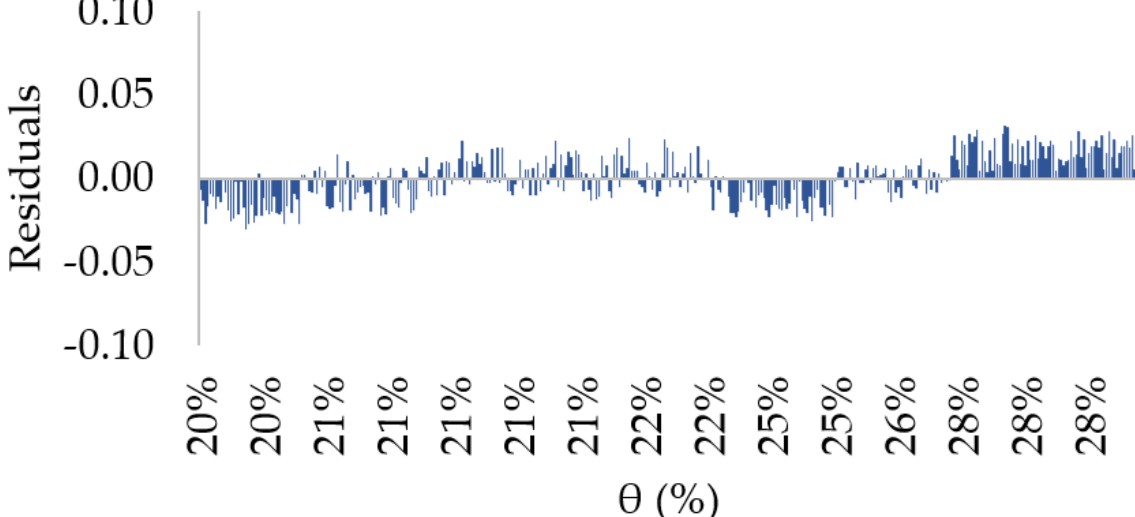

**Figure 15.** Residuals for combined class MLR1.

Regression steps were carried out to develop MLR2 (for BG class) and MLR3 (for VC class) in a similar fashion as described above for MLR1. Based on the regression statistics and model performance metrics, the model coefficients at step 1 of the iteration were accepted for both MLR2 and MLR3. All the final accepted independent variable coefficients and their corresponding SE for the three MLR models are presented in Table 7.

**Table 7.** MLR Model Parameters *.

| MLR Model | Class | Parameters | Intercept | R | G | B |
|---|---|---|---|---|---|---|
|  |  |  | $a_0$ | $a_1$ | $a_2$ | $a_3$ |
| MLR1 | VC + BG | Coefficients | **0.120659945** | **0.000339414** | **0** | **0.000374743** |
|  |  | SE | 0.0067 | 0.00006 | 0 | 0.000054 |
| MLR2 | BG | Coefficients | **0.28316194** | **−0.001057988** | **0.000911736** | **0.000169394** |
|  |  | SE | 0.012 | 0.000076 | 0.0001 | 0.000065 |
| MLR3 | VC | Coefficients | **0.199332456** | **−0.000451035** | **0.000407238** | **$6.87287 \times 10^{-5}$** |
|  |  | SE | 0.0040 | 0.00007 | 0.00005 | 0.00003 |

* Table Legend: Multiple Linear Regression (MLR), Standard Error (SE), Bare Ground (BG), Vegetation Covered (VC), Red (R), Green (G), and Blue (B).

The sites and the ROIs with the RGB data used to build MLR models are tabulated in Table 8. After averaging, the multiclass RGB data vectors boil down to seven instances, with three belonging to the bare ground class and four to the vegetation covered class. The MLR1 model's θ' predictions for the 370 RGB data vectors were grouped by their respective ROIs, and their averages were taken. MLR1 prediction results are presented in Table 8.

**Table 8.** MLR model data characteristics *.

| Site | ROI | RGB Rows Count | Class | Avg. Measured VSMC ($\theta$) | MLR1 Avg. Predicted VSMC ($\theta'$) |
|---|---|---|---|---|---|
| Metro | 1 | 44 | BG | 26% | 24% |
| Metro | 2 | 51 | BG | 25% | 25% |
| Terry | 2 | 74 | BG | 28% | 27% |
| Terry | 1 | 64 | VC | 21% | 21% |
| Sowell | 1 | 50 | VC | 22% | 21% |
| Sowell | 2 | 40 | VC | 20% | 21% |
| Sowell | 3 | 47 | VC | 21% | 21% |
| | | | | | RMSE: 0.01 |

* Table Legend: Bare Ground (BG), Region of Interest (ROI), Vegetation Covered (VC), Average (Avg.), Volumetric Soil Moisture Content (VSMC), and Root Mean Square Error (RMSE).

The averages of the predicted $\theta'$ were then compared with the average ground truth $\theta$ extracted from the respective ROIs. For instance, for the Metro Site ROI#1 location, the averages of 44 $\theta'$ predictions were taken and compared with the corresponding average measured $\theta$. The comparison between the averages of the predicted volumetric soil moisture content ($\theta'$) and measured $\theta$ for MLR1 after removing the results from outlier data points (Metro ROI#3 and Terry Road ROI#3) is presented in Figure 16. Although the coefficient of determination ($R^2$) was an excellent 0.94, it is not the best metric for measuring model prediction performance. Alternatively, MSE and RMSE are better prediction performance metrics, which are excellent for MLR1, as shown in Figure 16.

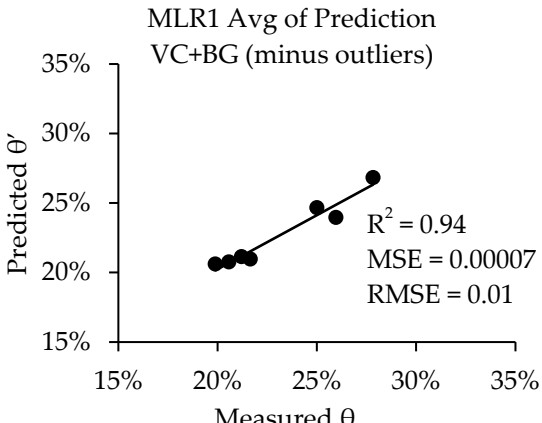

**Figure 16.** MLR1 average $\theta$ predicted vs. measured.

3.1.4. Comparative Analysis of RGB vs. $\theta$ Machine Learning Models

The performance of different regression machine learning models for predicting $\theta$ using RGB colors is compared using the widely accepted performance metrics: Mean Squared Error (MSE), Root Mean Squared Error (RMSE), and $R^2$. These are commonly used metrics for regression problems, which measure the average squared error between the predicted and actual values. The lower the MSE or RMSE, the better the model. R2 indicates the quality of predictions in relation to the actual measured ground truth data. R2 closer to 1 is indicative of excellent proximity between the measured and predicted $\theta$. Table 9 compares the different machine learning models predicting $\theta$ from RGB colors for the combined vegetation covered and bare ground classes. It is evident from the results that XGB1 performed the best, followed by MLR. SVR performance was the poorest among the three models evaluated in this study for predicting soil moisture content for MS highway slopes.

**Table 9.** Performance metrics of RGB vs. θ machine learning models.

|       | MSE (Test) | RMSE (Test) | $R^2$ |
|-------|------------|-------------|-------|
| XGB1  | 0.000084   | 0.009201598 | 0.9   |
| SVR1  | 0.00065    | 0.025509224 | 0.25  |
| MLR1  | 0.00007    | 0.01        | 0.9   |

### 3.2. Soil Moisture Characterization Using Diurnal LST Variation from Thermal Infrared Imagery

The diurnal LST variation is negatively correlated with θ [31]. While the vegetation-covered areas followed this relationship, the bare-ground areas did not consistently follow this logic. Therefore, the bare-ground LSTs were discarded and only the vegetation-covered data points were used to build θ vs. ΔLST curve fitting models.

ΔLST vs. Soil Moisture Content Regression Analysis Results

Diurnal surface temperature variance or ΔLST amplitude was calculated by taking the difference of LST values from the dawn and midday thermal images at the ROIs. Regression analysis was carried out to fit ΔLST vs. θ. The results from the linear and power curve regression analyses were compared, and the better-fitting power curve model was selected. The power curve model is presented in Figure 17.

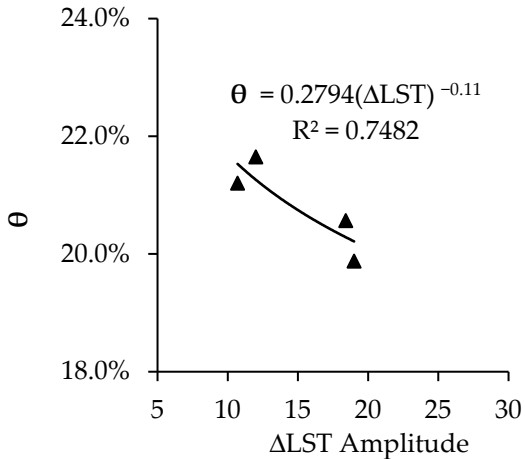

**Figure 17.** Vegetation-covered land Diurnal Surface Temperature Difference variation (ΔLST) vs. θ (power curve fit).

### 3.3. Validation

The generalizability of the θ prediction models was tested by performing validation tests on an unseen dataset. A non-instrumented slope near the university engineering facility in Jackson, MS, USA, was selected for validation. The site location along with the selected ROIs for validation (V1, V2, and V3) are presented in Figure 18. UAV flights were carried out to collect optical and thermal images of the validation site. The thermal images captured at dawn and midday are presented in Figure 19.

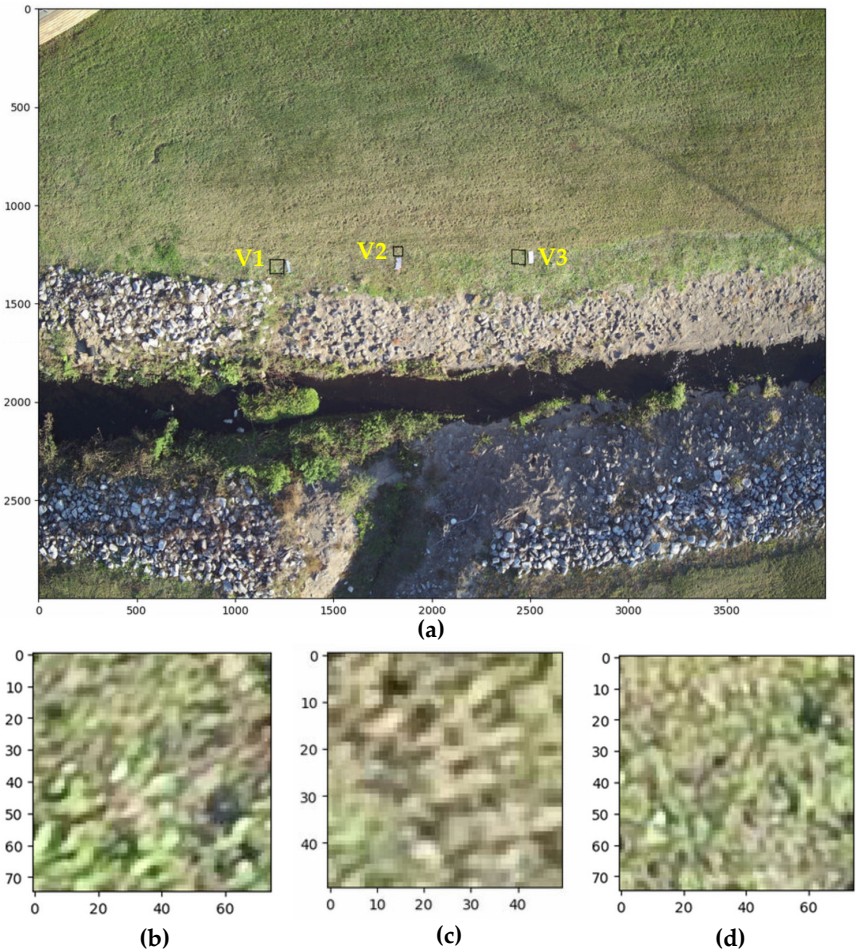

**Figure 18.** University slope. (**a**) UAV RGB image at 100 ft. altitude. (**b**) Region of interest V1. (**c**) Region of interest V2. (**d**) Region of interest V3.

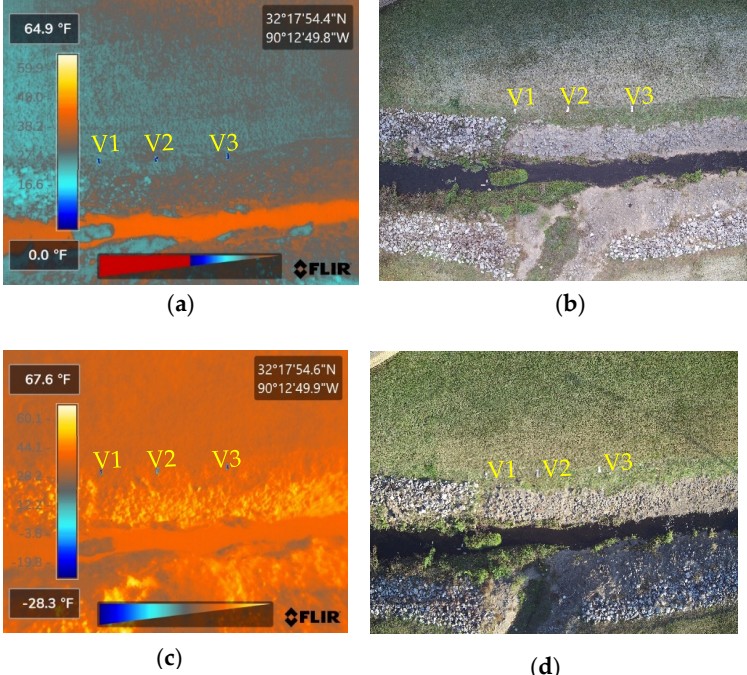

**Figure 19.** University slope UAV thermal and optical images at 100 ft. altitude. (**a**,**b**) Captured at dawn. (**c**,**d**) Captured at midday.

### 3.3.1. Validation of RGB vs. Soil Moisture Content Prediction Models

It is evident from the performance results in Section 3.1 that XGB and MLR outperformed SVR models in predicting θ. Therefore, only the XGB and MLR models were further validated by predicting θ from previously unseen RGB features.

XGB Validation

The average RGB values extracted from the unseen optical image were then fed to the trained XGB models and θ' was predicted. θ' was then compared with the ground truth θ, and the loss metrics were measured, namely MSE and RMSE. The results of the predictions by XGB models are presented in Table 10. All three XGB models performed well in predicting the θ'. The error in the prediction of XGB models is up to 10%, which is acceptable. These findings increase the confidence of the XGB models developed in this study to predict accurate near-surface soil moisture content.

**Table 10.** XGB model validation results.

| ROI | Class | Average RGB Values | | | θ | θ' | | |
|-----|-------|------|------|------|----------------|--------|--------|--------|
| | | R | G | B | Ground Truth | XGB1 | XGB2 | XGB3 |
| V1 | VC | 163.35 | 169.17 | 127.62 | 0.2368 | 0.2100 | 0.2473 | 0.2100 |
| V2 | VC+BG | 171.92 | 167.2 | 132.8 | 0.2301 | 0.2100 | 0.2272 | 0.2123 |
| V3 | VC | 167.09 | 169.75 | 128.125 | 0.2421 | 0.2100 | 0.2473 | 0.2100 |
| | | | | | MSE | 0.00072 | 0.00005 | 0.00067 |
| | | | | | RMSE | 0.02680 | 0.0069 | 0.026 |

MLR Validation

Depending on the ROI, the applicable MLR models of the respective classes were tested against the unseen dataset, and the results are presented in Table 11.

**Table 11.** MLR model validation results.

| ROI | Class | Average RGB Values | | | θ | MLR Predicted Results | | |
|-----|-------|------|------|------|----------------|--------|-------------|-------------|
| | | R | G | B | Ground Truth | Models | θ' | Residuals |
| V1 | VC | 163.35 | 169.17 | 127.62 | 0.2368 | MLR3 | 0.203319435 | 0.03301688 |
| V2 | VC + BG | 171.92 | 167.2 | 132.8 | 0.2301 | MLR1 | 0.22877787 | 0.00108936 |
| V3 | VC | 167.09 | 169.75 | 128.125 | 0.2421 | MLR3 | 0.201903469 | 0.03973272 |
| | | | | | | MSE | 0.0009 | |
| | | | | | | RMSE | 0.03 | |

### 3.3.2. Validation of ΔLST—Soil Moisture Content Regression Model

The ΔLST vs. θ linear regression model was further validated by testing the accuracy of θ prediction from unseen ΔLST values. Thermal images were captured at dawn and midday. All necessary calibration, verification, and optimization were performed precisely, as explained previously in Section 2.6.1. The land surface temperatures were then extracted from the three ROIs, V1, V2, and V3. Figure 19 shows the thermal image and the ROIs used to validate LST vs. soil moisture content models.

The power curve model with the equation $\theta = 0.2794(\Delta LST)^{-0.11}$ presented in Figure 17 was tested for its accuracy in predicting the soil moisture content. The prediction results are presented in Table 12. The observed results show that the power model predicted θ' with acceptable accuracy. The MSE and RMSE metrics are low, indicating that the soil moisture content model predictions are reliable.

**Table 12.** ΔLST vs. soil moisture regression model validation results.

| | ROI | LST Max (Midday) | LST Min (Dawn) | ΔLST | θ (Measured) | θ′ (Predicted) |
|---|---|---|---|---|---|---|
| Uni | V1 | 46.2 | 31.5 | 14.7 | 23.68% | 20.79% |
| Uni | V2 | 45.5 | 32 | 13.5 | 23.01% | 20.98% |
| Uni | V3 | 47.6 | 33.3 | 14.3 | 24% | 20.85% |
| | | | | | MSE | 0.0007 |
| | | | | | RMSE | 0.0273 |

## 4. Discussion

Soil moisture content variation has an undeniable influence on the stability of slopes [1–6], especially in the highway embankments of central Mississippi [7,40–43]. Therefore, continual monitoring of soil moisture variation in highway slopes and embankments is imperative. This issue is even more relevant now because the adverse effects of climatological loading on aging infrastructure have become more pronounced in recent decades. One of the effective ways to adapt to these changes is to frequently monitor the performance of infrastructure assets and take preventative measures prior to failures. Therefore, this study aimed to develop innovative soil moisture monitoring methods using UAV images and machine learning methods to aid in the predictive maintenance of highway slopes and embankments prone to frequent shallow slide failures.

Inferencing soil moisture using remote sensing data and machine learning algorithms has been implemented in agricultural hydrology applications [12,16,22,23,25]. However, these methods are rarely implemented for geotechnical infrastructure asset management. This study intended to fill the gap and develop near-surface soil moisture content prediction models using UAV images and machine learning methods.

Regression is the go-to method for forecasting or quantifying an underlying relationship between independent and dependent variables. The question is, "when is it prudent to use machine learning?" According to Chollet (2021) [44], machine learning should be avoided when all the rules, data, and underlying relationships are known, but only target answers are needed. Instead, the problem can be solved by logical and mathematical reasoning. However, when data and target answers are known, but the rules are missing, a machine learning method is more suitable to solve the problem. Therefore, machine learning approaches were employed to develop RGB vs. θ models, as the underlying patterns are unclear. Specifically, Support Vector Regression (SVR), Extreme Gradient Boosting (XGB), and Multiple Linear Regression (MLR) methods were evaluated for θ inferencing using RGB data.

On the other hand, since there is an underlying relationship between the soil surface thermal inertia and the near-surface soil moisture content, singe parameter linear and power curve fitting models were tested to infer θ from LST. All optical and thermal imagery data used in this paper were collected and analyzed by the authors.

The XGB models' performance metrics and the combination of parameters are presented in Table 5. XGB1 and XGB2 show excellent R2 scores compared to XGB3. The Explained Variance Regression Score (EVS) function of the sklearn library close to 1 represents a healthy prediction model. EVS for all three XGB models were >0.9, proving good prediction quality. Additionally, the test dataset's RMSE and MSE loss functions are close to zero, indicating well-performing models. Therefore, all three models are suitable for carrying out future predictions.

It is important to note that, although the SVR provided good results on the training dataset, the model failed to provide good predictions for the test dataset. Despite the good training results, the test results in terms of R2 score were poor. Furthermore, the test Explained Variance Regression Score (EVS) function is much farther from 1.0 and not indicative of healthy prediction models.

This paper is part of a more extensive study where the goal is to develop a methodology that can be adopted by transportation and engineering agencies to readily characterize and monitor soil properties, including soil moisture of highway embankments, through non-contact UAV surveys. Therefore, unaltered raw optical images in "as-is" status were used to extract RGB data. In future research, more parameters, including soil minerals, plant nutrition, and topography, can be incorporated to enhance nonlinear regression models for predicting the soil moisture content of geotechnical assets. This study can be expanded by including more datasets from different sites to eliminate bias from localized datasets. Furthermore, Multi-Level Perceptron (MLP or Neural Networks) and deep learning approaches can be explored to develop fusion models combining the color, thermal properties, and other parameters to predict soil moisture content from UAV images.

## 5. Conclusions

In this study, two different methods using UAV-captured optical and thermal images were developed to predict the near-surface soil moisture content of highway embankments in the Jackson, Mississippi, area. The first method used raw RGB color values from optical images to train machine learning models to infer θ. The second method used diurnal LST differences determined from thermal images to build regression models to estimate θ. The images were collected from three highway embankment sites in the Jackson, MS, metropolitan area. Bare Ground (BG), Vegetation Covered (VC), and mixed BG plus VC regions of interest were selected to collect ground truth soil moisture and extract RGB and land surface temperature values. Such multiclass ROIs were purposely selected to help generalize the θ inferencing capabilities of the developed models across both image classes.

This study conducted a literature review to identify frequently used machine learning models for predicting soil moisture content and compared their predictive performance to determine the best one. This comparative evaluation is necessary, as different models may perform differently depending on the problem and data. Three machine learning models were evaluated to predict soil moisture from RGB features. Extreme Gradient Boosting (XGB), Support Vector Regression (SVR), and Multiple Linear Regression (MLR) models were developed using ground truth soil moisture content as target and RGB pixel values as input features. The models were trained to predict θ from RGB values extracted from pixels belonging to Vegetation Covered (VC) and Bare Ground (BG) classes. The coefficient of determination, mean square error, and root mean square error metrics were used to evaluate the models' performances. The results showed that XGB and MLR outperformed SVR models in predicting soil moisture content, with each having an R2 score of >0.9 for predicting soil moisture. A smaller RMSE value indicates better performance of the model. In this case, the RMSE values for XGB, SVR, and MLR were 0.009, 0.025, and 0.01, respectively, for the test dataset, thus proving that the XGB model's performance was the best among the three models evaluated, followed by MLR. The XGB and MLR models were further validated by predicting soil moisture using previously unseen input data.

On the other hand, a power curve fit model was developed to predict the soil moisture content from thermal images. Radiometric data were first captured using a UAV-mounted FLIR Zenmuse-XT2 thermal sensor. After applying proper calibration, tuning, and validation steps, the land surface temperatures were extracted from the thermal images. A linear regression model was developed to predict θ from the diurnal variation of land surface temperature. The thermal inertia-based soil moisture prediction model ($\theta = 0.2794(\Delta LST)^{-0.11}$) provided better results for vegetation-covered ROIs than the bare-ground ROIs, with a coefficient of determination of 0.748.

The results of this study are promising and present an innovative, time-efficient, and non-contact method to monitor soil moisture variations within the shallow depths of highway embankments. Transportation and engineering agencies can adopt this methodology into geotechnical infrastructure asset management and predictive maintenance programs.

**Author Contributions:** Conceptualization, R.S., S.K., I.L.C. and F.A.; methodology, R.S. and M.N.; software, R.S. and O.E.A.; validation, R.S. and O.E.A.; formal analysis, R.S. and M.N.; investigation, R.S.; resources, S.K.; data curation, R.S.; writing—original draft preparation, R.S.; writing—review and editing, R.S. and M.N.; visualization, R.S. and M.N.; supervision, S.K.; project administration, S.K. and I.L.C.; funding acquisition, S.K., F.A. and I.L.C. All authors have read and agreed to the published version of the manuscript.

**Funding:** This research was funded by the Mississippi Department of Transportation (MDOT), grant number State Study 316.

**Data Availability Statement:** The data used to support the findings of this study are included within the article. The original details of the data presented in this study are available on request from the corresponding author.

**Acknowledgments:** The studies described in this paper are based on the work supported by the Mississippi Department of Transportation's (MDOT) State Study 316. The findings, conclusions, and recommendations expressed in this material are those of the authors and, necessarily, it does not reflect the viewpoints of the MDOT.

**Conflicts of Interest:** The authors declare no conflict of interest.

## Symbols and Abbreviations

| | |
|---|---|
| LST | Land Surface Temperature |
| UAV | Uncrewed/Unmanned Aerial Vehicle |
| SMC | Soil Moisture Content |
| $\theta$ | Volumetric Soil Moisture Content |
| $W$ | Gravimetric Soil Moisture Content |
| ROI | Region of Interest |
| BG | Bare Ground |
| ML | Machine Learning |
| SVR | Support Vector Regression |
| XGB | Extreme Gradient Boosting |
| MLR | Multiple Linear Regression |

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
