# Peer review of "Near-Surface Soil Moisture Characterization in Mississippi’s Highway Slopes Using Machine Learning Methods and UAV-Captured Infrared and Optical Images"

_remotesensing, doi:10.3390/rs15071888_

Round 1

Reviewer 1 Report (Previous Reviewer 2)

The author faithfully responded to all the opinions of the reviewers. In particular, the results and explanations of the accompanying figures were described in sufficient detail, and the purpose and direction of this paper were well represented through the revised discussion. The author's response and revised paper fully reflected all requests, so I believe that the current manuscript can be published.

Author Response

Reviewer 2 Report (Previous Reviewer 3)

Comments for remotesensing-2170615

The authors have replied all comments for last version, but the manuscript still has some questions.

1. Please check the equation (2), I think SwGs is a variable, but the authors may point three variables. This is not reasonable. In the last comments, I have pointed this question.

Author Response

Reviewer 3 Report (New Reviewer)

The research conducted by Salunke et al. is interesting and falls into the scope of the remote sensing journal. The manuscript can be further considered for publication after implementing minor revision aiming to improving the overall quality of the manuscript.

1-     The introduction is rather weak with regards to the application of ML models in the field. What are the previous research? What are the ML methods used to calculate the soil moisture content in the literature? And how your research contribute to the existing literature?

2-     Why did you use different ML models, not rather focusing on one of these methods?

3-     The implementation procedure of the ML models should be better presented to increase the reproducibility of the research.

4-     Bring a better discussion and comparison analysis about the performance of the ML models.

5-     Change the title of ‘conclusions’ to ‘summary and conclusions’.

Author Response

Reviewer 4 Report (New Reviewer)

Review on the paper “Near-Surface Soil Moisture Characterization in Mississippi's Highway Slopes Using Machine Learning Methods and UAV-Captured Infrared and Optical Images” by Salunke et al.

General:

The authors make an analysis of the soil moisture near highway slopes. This seems to be a relevant problem for road transport in Mississippi.

Now in detail

*Abstract

-Check line 32.

-The abstract must be improved by including some highlights of the results, quantitatively please.

*Keywords

-Too much keywords. 

*Introduction

I would like to know the importance of the work in the maintenance of the highway slopes. The civil engineers know how to construct roads by combining several sizes of rocks and materials. How much damaged places are in the area (for example)?

Line 74: if you write Uncrewed/Unmaned Aerial Vehicle (UAV), it is enough to improve the sentence.

The introduction must be improved by presenting some highlights of the main results.

However, it is well written and the number of references is correct.

*Materials and methods

Lines 126-131: Although you can use the full name and acronymous again, the paragraph is quite unclear. I recommend the use of the acronymous once defined.

 Lines 311-320: use bullets please

Check line 313 please

* Results

Table 4: explain all the parameters in the legend please.

Figure 13: the distribution of the points is quite unpleasant. Two separated clouds of the same variable and no data in the middle. I agree about the no ability of the SVR to predict, but I wonder if a simple previous plot would not be enough to avoid get into the complexities of the SVR.

Figure 14: Two separated clouds again. I wonder about the nature of the data because they are values of the same variable, but grouped! Can the authors explain that?

Lines 483 and Table 6: please, unify the notation

Table 6: Include the error in the estimation of the coefficients please. This is almost mandatory to evaluate which is relevant or not. Include the correlation please. The plot of the residuals is not enough.

Figure 17: it is not needed the presentation of two plots here. With the best one is enough. On the other hand, the fit is similar for me because the errors of the coefficients are not included and the regressions are useful in the data interval, perhaps the power model can be better out of them.

*Discussion

It is OK

*Conclusion

It is OK

Good luck!!!

Round 2

Reviewer 4 Report (New Reviewer)

Dear authors,

Thank you very much for your work and effort in clarifying my doubts.

I have an additional and important doubt.

When analyzing table 6, where you diligently included the standard deviation of the regression parameters, I have found that the errors :

MRL1 a2: the error is as high as the value of the parameter. So, a2 is meaningless and must be rejected.

MRL2: same with a2 and a3

MRL3: same with a3

These means that the MRL models must be improved in some some, for example changing the standard MLR for a stepwise regression or change the model.

In my opinion, this must be discussed and solved before publication.

Good luck!!!!

Author Response

Please see detailed response to the review comments in the attached file.

This manuscript is a resubmission of an earlier submission. The following is a list of the peer review reports and author responses from that submission.

Round 1

Reviewer 1 Report

Soil moisture detection with UAV sensors is an interesting topic. The authors presented their work on soil moisture content mapping for highway embankment safety. While the research topic is appealing, several major logical and structural flaws were found in this manuscript. (1) RGB and thermal sensors, how do you integrate these two types of data? The authors only report the RGB until section 5, and normally, this is when a paper finishes up with a conclusion. Soil moisture estimation using thermal images seems like an “add-on” study by the way you wrote, however, it seems that thermal images play a more important factor in soil moisture modeling. (2) The lack of transparency and logic makes this paper extremely difficult to follow, and the writing style can be improved, an example is a lot of repeated paragraphs (e.g., lines 187 -197). (3) A manuscript of 10 sections is very uncommon, I suggest the authors do substantial learning and literature review before writing a paper. Better work on the references is required.  (4) My final thought is that XGB is not necessary given the factors you model for the soil moisture are not very complicated, linear regression is good enough if your data is of high quality.

Given these major flaws, I am very sorry to report that acceptance for publication is not recommended for this manuscript.

Specific comments/revision suggestions:

Figure 6 to figure 10: the RGB data were reflectance or raw data?

Line 169-175 please rewrite this paragraph. The discussions here are not legit to appear in the method section. Break them either into the introduction or discussion section.

Line 187 -197 content repeated.

Line 176-185  writing style problem: too many tech details. These details are good for engineering reports, not meant for a research article.

Line 198 “Jupyter notebook and Python script were used to crop the…” the use of this sentence as well as repeating it several times is not necessary for a research article.

Figure 3: you must specify the image resolution. They are very blurry.

Figure 4, why Region of interest 1? It’s covered by vegetation.

Why don’t sample your region along the road, since they are road embankments?

Figure 7,8,9 violates your assumption, “larger SMC is darker”.

Section 5: Thermal is a new section? You should bring up what your research question upfront, not as an add-on study.

XGB Results: I reserve my comments on the necessity of XGB. When data is high quality, linear regression is enough.

Reviewer 2 Report

1. Review

In this paper, UAV data is used to predict soil moisture in a highway embankment. In addition, machine learning algorithms were utilized to predict soil moisture. The methodology used in this paper provides an easy way to characterize and monitor soil moisture. However, some modifications are required for publication of the thesis.

2. comments

1) In Section 2, it was mentioned that a specific gravity of 2.7 was considered to derive VSMC. I wonder if the reason for using the specific gravity of 2.7 here is the same as the reason mentioned in Section 3.2.

2) The spatial extent of the UAV data is not presented. Something like this needs to be added to the text. Also, in figures such as Figure 3, the meaning of the x-axis and y-axis is ambiguous.

3) In section 3.2, it is only mentioned that GSMC was determined according to the procedure described in ASTM (2010) standard D2216, but a brief explanation needs to be added for the reader's understanding.

4) The phrase "when is it prudent to use machine learning?" in section 3.3 is interesting. If this was mentioned in the discussion, I think it would explain why the author used machine learning algorithms.

5) In Section 4.1, outliers were removed from the data to prevent errors. However, the results were derived using data with outliers. Is there a reason you used data with outliers? If not, it would be better to derive results using only data with outliers removed.

6) In Section 5.1 the methodology for temperature calibration, validation and optimization is well described, but nothing about the results appears. I think it is necessary to present the results using tables or figures.

7) In Section 8, two verification methods are presented. In this text, the RGB verification method is judged to be applied only to the XGB model, and the thermal infrared image verification method is judged to be applied only to the multiple linear regression model. I wonder why the RGB verification method was applied only to the XGB model, and the thermal infrared image verification method was applied only to the multiple linear regression model. Also, I wonder if the SVR model has not been validated, and if the validation has not been performed, is there a reason for that? Please add this information to the text.

Reviewer 3 Report

General comments:

The study focused on developing soil moisture prediction models from the color and temperature information embedded in the optical and thermal images captured by an Unmanned Aerial Vehicle (UAV). Constructed nonlinear regression models to predict soil moisture content using pixle RGB values. The study is suitable for published in this journal. Overall, I have some minor comments and suggest accept after minor revision.

Specific comments:

 1. Line 35 in abstract: the full name of SMC should be given before the abbreviation.

2. Please check the citation rules, the citation used in lines 91 and 92 may be not suitable.

3. Check equation (1), what are the meanings of “SwGs” and “wGs”? Please give.

4. Figure 5 is same to Figure 4, I think the authors gives a wrong picture in Figure 5, it should be the picture of “Sowell Road Slope UAV RGB Image at Dawn”.

5. Section 6.3, in lines 431 to 441, what many soil moisture content observations are used to construct the MLR? What many to predict? In figure 13, the observations are not too many.